# Green Antimicrobials as Therapeutic Agents for Diabetic Foot Ulcers

**DOI:** 10.3390/antibiotics12030467

**Published:** 2023-02-25

**Authors:** Ines D. Teixeira, Eugenia Carvalho, Ermelindo C. Leal

**Affiliations:** 1Center for Neuroscience and Cell Biology, Center for Innovative Biomedicine and Biotechnology, University of Coimbra, Rua Larga, 3004-504 Coimbra, Portugal; 2Institute for Interdisciplinary Research, University of Coimbra, 3004-504 Coimbra, Portugal

**Keywords:** diabetic foot ulcer, antimicrobials peptides, green synthesis, environmentally friendly

## Abstract

Diabetic foot ulcers (DFU) are one of the most serious and devastating complications of diabetes and account for a significant decrease in quality of life and costly healthcare expenses worldwide. This condition affects around 15% of diabetic patients and is one of the leading causes of lower limb amputations. DFUs generally present poor clinical outcomes, mainly due to the impaired healing process and the elevated risk of microbial infections which leads to tissue damage. Nowadays, antimicrobial resistance poses a rising threat to global health, thus hampering DFU treatment and care. Faced with this reality, it is pivotal to find greener and less environmentally impactful alternatives for fighting these resistant microbes. Antimicrobial peptides are small molecules that play a crucial role in the innate immune system of the host and can be found in nature. Some of these molecules have shown broad-spectrum antimicrobial properties and wound-healing activity, making them good potential therapeutic compounds to treat DFUs. This review aims to describe antimicrobial peptides derived from green, eco-friendly processes that can be used as potential therapeutic compounds to treat DFUs, thereby granting a better quality of life to patients and their families while protecting our fundamental bio-resources.

## 1. Introduction

Diabetes mellitus (DM) is a major public health problem with rising prevalence worldwide [1,2]. In 2021, approximately 537 million adults were diagnosed with DM, and this number has been predicted to rise to over 783 million by 2045 [1]. Along with the rising prevalence of diabetes, DM-associated complications are also expected to increase, resulting in high morbidity, mortality, and health expenditure rates, due to the required hospitalizations and specialized care [1,2]. Diabetes can lead to life-threatening and disabling health complications such as retinopathy, neuropathy, cardiovascular diseases, nephropathy, and diabetic foot ulcerations [1,2].

Diabetic foot ulcers (DFUs) are one of the most prevalent complications of diabetes, and are, in part, are associated with peripheral vascular disease and peripheral neuropathy [2]. It is estimated that 19–34% of patients with diabetes will develop DFUs in their lifetime [3]. Approximately 85% of these patients undergo non-traumatic lower limb amputation [2,3,4,5].

Wound healing is a dynamic and complex biological process aimed to restore skin function after trauma. Nevertheless, under diabetic conditions, this process is impaired [2,6,7]. Diabetic individuals exhibit multiple risk factors such as hyperglycaemia, prolonged hypoxia, chronic inflammation, peripheral neuropathy, impaired neovascularization, and difficulty fighting infections that compromise and delay the normal wound healing process [2]. This results in the occurrence of chronic non-healing wounds in a persistent pro-inflammatory state [6,7]. Moreover, about 60% of DFUs become infected with bacterial colonies, contributing to chronic wound healing failure [2,6,7].

As a result of diabetic foot infections (DFI), patients are frequently hospitalized and receive multiple antibiotic courses over the sequence of treatments [8]. Today, the emergence and spread of multi-drug resistant bacteria impose a rising threat to global health [8,9]. For this reason, it is pivotal to find green and less environmentally harmful alternatives for fighting these resistant microbials.

Antimicrobial peptides (AMPs) are conserved bioactive molecules of the innate and adaptative immune system which can be found in all forms of life [2,8,9]. Recent studies have demonstrated that these short peptides, composed of 15 to 60 amino acids, have various mechanisms of action that play important roles in the fight against infection by providing broad-spectrum antimicrobial activity against Gram-negative and Gram-positive bacteria, fungi, and viruses, as well as in wound healing [2,8,9]. Nevertheless, these small peptides can present some limitations such as low structure stability, high cytotoxicity, low hydrosolubility, salt sensitivity, and poor selectivity, as well as high production costs [2,8,9].

Direct eradication of microorganisms can be achieved by AMPs through mechanisms such as membrane disruption, interaction with intracellular targets, recruitment, and activation of immune cells [2]. These molecules can also promote wound healing by re-epithelization, support of angiogenesis, and enhancement of extracellular matrix synthesis [2,8]. Thus, AMP-based approaches may be a good solution to fight the emergence of antimicrobial resistance [2].

With the increasing awareness of climate change and high pollution levels, there has been a heightened desire to protect the environment. To address this issue, scientists have been focusing on the development of innovative and environmentally friendly techniques for AMPs extraction and synthesis. The use of plants and microalgae as sources of novel AMPs has been considered to be sustainable and very promising. However, more studies are needed in order to improve the current protocols in use.

Therefore, this review aims to describe AMPs derived from greener, eco-friendlier processes that can be used as potential therapeutic compounds to treat infection, including DFUs, thereby granting a better quality of life to patients and their families while protecting our fundamental bio-resources the environment.

## 2. Diabetic Foot Ulcers

Wound healing is a dynamic and complex biological process aimed to restore skin function after trauma [6]. This process is achieved through four successive phases with limited overlap: hemostasis, inflammation, proliferation, and remodeling [2,6].

The hemostasis phase initiates immediately after skin injury and is characterized by the constriction of the damaged blood vessels and the activation of platelets. Platelet aggregations will promote the formation of a fibrin clot that seals the damaged endothelium, stopping the bleeding [2,6]. Once the bleeding is controlled, the inflammatory phase begins. Damaged cells release chemokines and cytokines, which recruit inflammatory cells to the site of injury. Neutrophils are the first cells mobilized to the wound site and are responsible for microbial clearance [2,6,7]. These cells are then followed by pro-inflammatory macrophages (M1), which induce the clearing of apoptotic cells, including neutrophils, and secrete growth factors and cytokines that promote the inflammatory response by attracting adaptive immune system cells to the site of injury [2,6,7]. As the inflammatory stage is resolved, these cells undergo a phenotypic alteration to a reparative state (M2) that promotes tissue regeneration [7]. The proliferative phase begins once the inflammation decreases. This phase is characterized by the formation of granulation tissue, contraction of the wound edges, re-epithelization, and neovascularization [2,6,10]. The final step of wound healing is the remodeling phase, which includes collagen fiber reorganization, remodeling, and maturation of the scar tissue, as well as an increase in its tensile strength [2,6,10].

For a wound to heal successfully, all four phases need to be very well coordinated in the right order and within the appropriate time period [2,6]. Nevertheless, under diabetic conditions, this process is impaired resulting in the occurrence of chronic non-healing wounds in a state of persistent pro-inflammation [2,6,7]. This is characterized by an accumulation of immune cells, an increase in the M1/M2 macrophage ratio, increased generation of reactive oxygen species (ROS), and pro-inflammatory cytokines [6]. Moreover, about 60% of DFUs develop bacterial infections which contribute to the failure to heal chronic wounds [2,6].

Patients with DFIs are regularly hospitalized and are frequently exposed to numerous courses of antibiotics [8,9,11]. Today, the emergence and spread of bacteria that are resistant to conventional antibiotics, including those used within the hospital environment, impose a rising threat to global health [9,11]. This problem greatly hampers DFU treatment. For this reason, the development of alternative compounds with the capacity to downregulate the inflammatory response and control pathogen infection is urgently required.

## 3. Microbiota in Diabetic Foot Ulcers

The skin is the largest organ in the human body, and it has an essential role as a multifunctional barrier, protecting our body against pathogens or toxic substances [12]. The human skin contains a large and diverse composition of living microorganisms known as the skin microbiota. Most of these microorganisms are harmless and even advantageous to their host, protecting against invasion by more pathogenic organisms and contributing to skin homeostasis. However, the disruption in this balanced microbiota system can enhance the susceptibility to skin disorders, including infections [12,13].

The physical and chemical characteristics of the skin surface may vary according to environmental and specific host factors. These features can influence colonization by the skin microbiota and determine unique sets of microorganisms [12]. Under diabetic conditions, skin integrity is compromised by several factors, contributing to impaired tissue regeneration and alterations in the local skin microbiome [14,15]. Persistent hyperglycemia creates an excessive nutrient source for microbes and reduces innate immunity by causing poor chemotaxis, phagocytosis, and cleansing of pathogens by neutrophils [15]. Moreover, peripheral vascular disease hampers the action of the host’s immune response, in part due to reduced blood flow [15]. Peripheral neuropathy aggravates minor traumas and increases forefoot pressure, facilitating the entry of microbes [14,15]. All these factors contribute to microbial colonization, biofilm formation, and clinical infections that impair wound healing and contribute to serious complications including osteomyelitis and lower limb amputation [14,15]. Approximately 60% of all DFU cases are estimated to develop DFI [2].

The DFU microbiota has been extensively studied [8,15,16,17,18]. A longitudinal study of patients with DFU (*n* = 100) used shotgun metagenomic sequencing to profile chronic wound microbiota and investigated its role in clinical outcomes and the response to therapy. The majority of bacteria present in diabetic foot ulcers were Gram-positive strains, such as *Staphylococcus aureus,* methicillin-resistant *S.aureus* (MRSA), and *Corynebacterium striatum*, as well as Gram-negative bacteria, including *Pseudomonas aeruginosa* and *Alcaligenes faecalis* [15]. These results were in accordance with other microbial studies that also include *Escherichia coli* and *Proteus* spp. as the most predominant isolated strains from DFU [11,15,16]. Moreover, some anaerobes have also been identified in the deep tissue within diabetic wounds, including *Peptostreptococcus* spp., *Bacteroides* spp., *Prevotella* spp., and *Clostridium* spp. [2,8,15].

The majority of DFI are polymicrobial in nature, and mixed microorganisms, including fungi, are frequently prevalent. The prevalence of pathogenic fungal species and subsequent mycotic infections are responsible for an increased risk of diabetic foot syndrome development and poor clinical outcomes [2,8,15,16,17]. The most commonly isolated fungi are *Candida* spp., *Trichophyton* spp., *Aspergillus* spp., *Trichosporon* spp., and *Cladosporium herbarum* [2,15,16,17].

DFUs are considered polymicrobial ecosystems composed of highly dynamic and diverse microbial communities [8,18]. These microorganisms can exist independently or can organize into functionally equivalent pathogroups (FEP), where commensal and pathogenic bacteria co-aggregate symbiotically in a self-produced protective polysaccharide matrix with transformed phenotype known as biofilms [2,18]. Since biofilms hamper local access to antimicrobial agents and the host’s immune system, the wound healing becomes stalled and infection is very difficult to resolve, further promoting chronic infected wounds [2,8,18]. A prospective study revealed that approximately 97.6% of microbial isolates from chronic DFUs were multi-drug resistant (MDR) with 46.3% of MDR isolates having the ability to form biofilms. *Staphylococcus aureus* is the biofilm forming most predominant strain [19].

## 4. Antimicrobial Peptides

To combat multi-drug-resistant bacterial infections, there is a clear need for the development of novel alternative compounds and therapeutic strategies. In recent years, increasing attention has been paid to AMPs as a novel class of antimicrobials with great clinical potential [9].

Antimicrobial peptides, also known as host defense peptides, are effector molecules of the innate immune system which can be found in all living organisms [9,11,20]. These small molecules play important roles in fighting infection through broad-spectrum antimicrobial activity, host’s immunomodulatory functions, as well as other functions relevant to wound healing (Figure 1) [13].

There is a large diversity of known AMPs: more than 5000 antimicrobial peptides have been characterized and synthesized, and this number is expected to increase in the coming years [21]. Based on their structure, AMPs can be classified into four different groups: α-helical, extended, β-sheet, and cyclic [9,20,22]. Their secondary structures provide each peptide with a functional specificity [20].

Upon injury, the innate immune system recognizes pathogen-associated molecular patterns (PAMPs), including lipopolysaccharides (LPS) [2]. This, in turn, leads to the production of AMPs by skin resident cells, such as keratinocytes, and by infiltrating leukocytes, circulating neutrophils, and tissue macrophages [22,23]. As a result of the overexpression of these small molecules, the body is able to respond to injury and infection quickly and effectively [2,9,22].

The most frequent mechanism of action of AMPs consists of targeting bacterial cell membranes directly [2,9]. AMPs often include positively charged residues and multiple hydrophobic residues, which allow them to settle electrostatic interactions between their cationic membrane and the anionic bacterial membranes [9]. As a result, the pathogenic bacterial membrane is disrupted, often through the formation of pores. This subsequently leads to the insertion of AMPs into the membrane, causing bacterial cell death [2,9]. Additionally, it has also been described that some AMPs have the ability to cross bacterial cell membranes without affecting their integrity. These peptides inhibit essential bacterial intracellular functions such as nucleic acid and protein synthesis, as well as enzyme activity, thus leading to cell death [24].

In addition to the direct eradication of microbes, AMPs can have an indirect antimicrobial effect by modulating and enhancing the hosts’ adaptative immune responses [2,13,23]. These peptides can act as chemoattractants, recruiting and activating immune cells. This leads to the increasing expression of proinflammatory cytokines, thereby suppressing potentially harmful inflammation [2,9,20,22]. AMPs are also able to promote wound healing by the induction of cell migration, proliferation and differentiation, re-epithelization, support of angiogenesis, and enhancement of extracellular matrix synthesis [2,13].

In human skin, cathelicidins and defensins are the most prevalent classes of endogenous AMPs, particularly cathelicidin LL-37 and human β-defensins (hBDs 1–3) [22,23].

It is important to note, however, that under diabetic conditions, the expression and/or activity level of endogenous AMPs may be greatly affected. This in turn may contribute to the inadequate infection control and impaired wound healing often observed in the presence of diabetes [2]. Gonzalez-Curiel et al. have demonstrated increased susceptibility to infectious diseases in patients with type 2 diabetes due to lower levels of CAMP (LL-37) and DEFB4 (hBD-2) genes in peripheral blood cells [25]. Moreover, Al-Shibly et al. showed that although DFUs may express some endogenous AMPs, their expression levels are inefficient to suppress secondary infections and promote wound healing [26].

Therefore, in order to maximize the activity of these peptides at the wound site and enhance wound healing, it is necessary to maintain or increase their expression and activity levels as well as overcome some of their limitations. The use of chemical modifications could be considered a potential strategy for increasing the stability of these AMPs within the DFU microenvironment, decreasing their toxicity, and improving their antimicrobial and wound-healing functions [2].

## 5. Green and Eco-Friendly Processes to Obtain AMPs

With the growth of industrialization, environmental concerns began to emerge [27]. At the beginning of the 1990s, Paul Anastas and John Warner proposed the twelve principles of green chemistry (Figure 2) [27,28]. These rely on reducing or removing the hazardous substances from the manufacture process, the application of chemical products, and avoiding a generation of toxic secondary products and waste from these processes [27,29]. In accordance with these principles, environmentally friendly actions should be developed at every stage of the product’s lifecycle, from conception to its synthesis, processing, analysis, and disposal [27]. With this in mind, green chemistry aims to minimize threats to human health and the environment by reducing or eliminating the use and production of toxic compounds [27,29].

In recent years, the awareness of climate change and the high levels of pollution has increased the consciousness of our ecological responsibilities. Pharmaceutical industries are a major contributor to climate change and environmental pollution worldwide [30]. Pharmaceutical residues, such as antibiotics, are one of the main anthropogenic pollutants [31]. These are associated with the increased selection of resistant pathogens that are present in the environment. With the excessive use of antibiotics in medical practice, veterinary medicine, and agriculture worldwide, antibiotic-resistant bacteria and antibiotic-resistance genes (ARGs) are becoming increasingly prevalent, harming human, animal, and environmental health [32]. Faced with this reality, it is pivotal to find greener and less environmentally harmful alternatives to help fight these resistant pathogens.

The increasing emphasis on green chemistry has led chemical-pharmaceutical industries and laboratories to concentrate on diminishing their environmental footprint [29,33].

### 5.1. Green Extraction Methods to Obtain AMPs

To date, several thousand AMPs have been isolated from different natural sources [20,34,35]. However, the techniques used to isolate these small bioactive molecules have not always been the most environmentally friendly. The conventional extraction methods have contained numerous toxic and flammable solvents, used high temperatures and energy demands, and extended extraction times that lead to molecule degradation [36,37]. The implementation of green chemistry in extraction processes is essential to reduce the release of chemicals hazardous to human health and the environment [36]. Extraction methods based on green principles consume less energy, exploit alternative solvents, and utilize sustainable resources while producing high-quality and safe extracts [36,37].

Plants and algae are considered two promising sustainable sources of a wide diversity of natural AMPs [34,35,37]. A variety of methods can be used to extract bioactive molecules from plant-based material. The general approach often includes three main stages: plant material homogenization, extraction, and purification [38]. Some traditional extraction techniques include maceration, hydrodistillation, and Soxhlet extraction [37]. However, many approaches such as ultrasound-assisted extraction (UAE), pressurized liquid extraction (PLE), pulsed electric fields extraction (PEFE), microwave-assisted extraction (MAE), and supercritical fluid extraction (SFE), have been developed and optimized to ensure greater sustainability [36,37].

In ultrasound-assisted extraction, sound waves and frequencies are used to induce the rupture of cell walls and release their content [39,40,41]. This technique has significant benefits such as shorter extraction time, lower solvent and energy depletion, and greater extraction yield [39,40,41]. PLE is a solid–liquid method that involves the application of high pressure and temperature. This results in an increase in the solvent’s boiling point, promoting its quick infiltration into the sample matrix. Due to its high extraction temperatures, this method is not recommended for compounds that show heat sensitivity [40]. The pulsed electric field extraction (PEFE) method uses electromechanical forces to create irreversible electroporation in the cell membrane, which leads to its increased permeabilization and subsequently enhances the mass transfer rate [41]. Another emerging clean technology is the MAE. The basic principle of this methodology is the transformation of electromagnetic energy into thermal energy. This leads to the heating of the intracellular moisture, causing evaporation, and, consequently, an increase in intracellular pressure that results in both organelle and cell wall disruption [41]. SFE is an advanced separation technique that employs natural chemical components, such as supercritical fluid CO_2_, as solvents [36,37,39]. This innovative technique can be applied to both plant and algae-based materials [36]. SFE represents a sustainable alternative to traditional solvent extraction methods, thus granting better efficacy and selectivity, higher diffusivity, with reducing extraction time [36,37,39]. Furthermore, supercritical fluids can be recycled and reused, resulting in a decrease of waste generation [37].

Microalgal AMPs can be obtained by solvent extraction, microbial fermentation, or enzymatic hydrolysis, with the latter being the most used extraction technique [42,43,44]. Sun et al. isolated the antibacterial peptide SP-1 from *Spirulina platensis* through enzymatic hydrolysis. This peptide showed considerable antibacterial effects against *Escherichia coli* and *Staphylococcus aureus* [45]. Further studies demonstrated antioxidant, antihypertensive, anti-diabetes, and anti-obesity properties of bioactive peptides derived from the same microalgae [46]. In another study, the peptide KLENCNYAVELGK was extracted from pepsin hydrolysates derived from *Limnospira sp*. This bioactive molecule also exhibited antibacterial activity against *Escherichia coli* and *Staphylococcus aureus* [47].

The enzymatic hydrolysis extraction method allows the isolation of bioactive peptides using certain commercial enzymes and physicochemical conditions such as optimal temperatures (below 100 °C) and pHs (close to neutral) [43,46]. In order for an enzyme to perform its functions, it must first bind to the substrate and then proceed with the catalysis. In this process, an enzyme membrane reactor system is used to hydrolyze the extracted proteins [43,44]. The membrane filter retains large particles that are then recycled back into the hydrolysis tank, only allowing hydrolyzed and small fractions to pass [43]. The purification of the obtained peptides can be achieved by various chromatographic methods such as ion exchange, reverse-phase high-performance liquid chromatography (RP-HPLC), and gel chromatography [46]. The purification method is selected based on the physical and chemical characteristics of the bioactive peptides [43]. Finally, to characterize the bioactive peptides at a molecular level, spectrophotometric techniques, such as liquid chromatography-mass spectrophotometry (LC–MS) and mass-mass spectrophotometry (MS–MS) can be used [44,46].

Enzymatic hydrolysis presents some advantages when compared to other techniques. This method has shown an improved reaction rate, higher yields, ease to scale-up, and high specificity, thus allowing the obtention of AMPs with the desired molecular size-weight [43,44,46]. It is considered an environmentally friendly process since it usually does not generate by-products or use any synthetic chemical agents [43]. Moreover, the mild temperatures and pH conditions under which this technique is performed allow for lower operating costs and energy demands [43,44].

In addition to the development of greener extraction methods, other purification techniques have been reviewed and improved in order to reduce their ecological footprint. Recent methodologies include mixed-mode chromatography (MMC) columns, in which the separation is characterized by multiple interactions between the solute and the stationary phase [48]. Another technique is multicolumn counter-current solvent gradient purification (MCSGP). This technology represents a chromatography purification process that allows the separation of different components by countercurrent movements between the stationary and mobile phase. Compared with standard methods, these recent techniques (MMC and MCSGP) exhibit higher yields and faster separation rates while being able to decrease water consumption and the generation of toxic waste. Moreover, supercritical fluid chromatography (SFC) has shown to be a promising and eco-friendly method compared to HPLC. SFC presents several advantages, such as the use of less organic solvents and toxic modifiers and the possibility of recycling the CO_2_ (used as mobile phase) during purification. It is also noteworthy that this process reduces the organic waste to less than one third in comparison with the current methods [48].

### 5.2. Green Synthesis of AMPs

So far, numerous AMPs have been discovered and isolated from various natural sources. Over the past years, several studies have been focused on unravelling the promising therapeutic properties of these bioactive molecules [2]. However, most of these small peptides possess some inherent limiting characteristics, such as low hydrosolubility, low stability, salt sensitivity, poor selectivity, and sometimes high cytotoxicity to the host [2,8,9,49,50,51].

To solve the hurdles presented by some natural AMPs, scientists began the development of new and improved synthetic antimicrobial peptides (SAMPs). These new molecules use the natural AMPs amino acid (aa) sequences as templates but undergo some specific chemical alterations, new formulations, or fusion with various synthetic elements in order to enhance their properties [49,50,51]. In comparison with natural AMPs, SAMPs exhibit improved effectiveness, reduced cytotoxicity, and greater resistance to protease degradation [49,50]. These characteristics improve their therapeutic potential in clinical applications [2,50].

In the development of synthetic peptides, a variety of approaches have been used [2,50,51]. One of the main strategies consists of chemical modifications. These approaches are characterized by punctual alterations (substitution, deletion, or addition of new aa) in the amino acid sequence of the active sites of naturally occurring AMPs, resulting in semi-synthetic antimicrobial peptides [49,51]. Different modification approaches revealed significant advantages in AMP manipulation: glycosylation (glycan is covalently attached to the peptide) [52]; lipidation (lipid group is covalently attached to the peptide) [2,53,54]; hydrazidation (attachment of a hydrazide to the peptide) [2,53,54].; guanidination (lysine residues are converted into homoarginine residues) [2,54]; and small molecule conjugation (small molecules such as antibiotics, peptides, and others, are incorporated into the AMP structure) [2,54].

Chemical synthesis has quickly become the most used method for obtaining synthetic AMPs [51,55]. The solid-phase peptide synthesis (SPPS) is widely used as the standard technique to obtain small or medium-sized peptides (30 to 50 residues). In fact, it is demonstrated that this method improved potency, reliability, and speed [50,51,56]. Zapotoczna et al. have demonstrated the antibacterial effectiveness of seven synthetic AMPs (D-Bac8c, WMR, HB43, P18, Omiganan, Polyphemusin, and Ranalexin) against *Staphylococcus aureus* biofilm infections. Among these SAMPs, D-Bac8c has shown higher potency in microbial eradication of *S. aureus* biofilm infections in both in vitro and in vivo studies [57]. In the SPPS process, successive and protected amino acids are added to a growing peptide chain, anchored to a solid support (usually resin). In addition, it involves several deprotection and washing steps to eliminate solvent waste [55,58]. The generation and accumulation of by-products are directly affected by the sequence and length of the desired peptide as well as the choice of resin used in this process, thereby affecting the efficiency of the peptide synthesis [59].

Despite the significant achievements of chemical synthesis, this process has a huge negative environmental impact. It is particularly responsible for enormous amounts of hazardous waste [58]. Moreover, this technique requires high consumption of toxic solvents, such as N,N-Dimethylformamide (DMF), dichloromethane (DCM), and *N*-methyl-2-pyrrolidone (NMP) [58,59,60]. In fact, some studies associated exposure to these compounds with hepatotoxicity and chronic toxicity [58]. The rising health and environmental concerns prompted the scientific community to develop greener and environmentally friendly alternative approaches for peptide synthesis [58,59,60]. Overall, several attempts have provided some promising results while granting high yields of peptides [58].

Most efforts have been made regarding greener and safer solvents. In 2009, Declerck et al. reported the first successful solvent-free peptide synthesis by the ball-milling technique [61]. Later, an in-depth study by Lopez et al. found that the best suitable replacement for the toxic compound DMF in SPPS was the green solvent *N*-butylpyrrolidinone (NBP) [60]. More recently, Jadhav et al. revealed that varying the composition of green binary solvent mixtures resulted in the mitigation of SPPS side reactions [59].

The use of greener resins is also another important research field currently in development [48,62,63]. New eco-friendly resins, such as polyethylene glycol-based, are able to swell to a significant extent in the majority of solvents and show great compatibility with new green solvents [48]. Recently, another greener resin derived from renewable resources, poly-ε-lysine resin SpheriTide, has proven to be biodegradable and to have a high loading capacity [48,63].

Over the last decades, green analytical methodologies have been introduced in order to make experimental protocols even safer and more environmentally benign [64,65,66]. In this context, green chromatographic methods have been given increasing importance [64]. The primary goal of green chromatography is to make the process of analysis greener at every stage by replacing existing solvents with greener alternatives, reducing solvent use, and decreasing waste generation [64,65]. In liquid chromatography (LC), the solvent reduction can be achieved by several strategies such as decreasing the column length with small particle size, using ultra-high performance liquid chromatography systems (short columns with reduced internal diameters), microflow and capillary HPLC columns, or even through the use of elevated temperatures [64]. Moreover, several green LC methodologies already implement the use of greener solvents (Brij 35, sodium dodecyl sulfate, and propylene carbonate) as a substitute for toxic solvents [64,65]. For example, replacing acetonitrile with ethanol (a nontoxic alternative) is a good strategy to improve LC techniques [64]. Additionally, as referred before, several processes can be optimized by the implementation of greener solvents and materials/instruments similar to resins.

### 5.3. Green and Sustainable Sources of AMPs

#### 5.3.1. Plants

As plants have evolved, they have developed refined defense mechanisms that enable them to protect themselves from potentially harmful organisms [34]. These include chemical barriers, in which plants produce a high number of toxic defense molecules, including AMPs [34,67]. In response to pathogen stimulation, multiple AMPs can be produced from different parts of the plant such as roots, seeds, flowers, fruits, stems, and leaves, or even from the whole plant [67]. A single plant species can contain numerous AMPs with various functional characteristics, structures, different expression patterns, and particular targets [34,67]. Based on their structure, plant AMPs can be classified into eight different families: thionins, defensins, hevein-like peptides, knotting-type peptides, α-hairpinin, lipid transfer proteins, cyclotides, and snakins [34,67].

Further research has revealed that plant AMPs have many other physiological functions in addition to their antimicrobial roles. These include the regulation of plant growth and development, as well as the ability to treat numerous diseases effectively [34]. Therefore, natural peptide extraction is very important since these molecules represent a promising alternative to conventional antibiotics with broad potential applications [34,38,67].

In 1970, Okada and Yoshizumi isolated the first AMP of plant origin, the purothionins, from barley (*Hordeum vulgare*) endosperm. These peptides showed antimicrobial effects against *Pseudomonas solanacearum, Erwinia amylovora, Xanthomonas phaseoli,* and *X. campestris, Corynebacterium poinsettiae, C. fascians, C. flaccumfaciens,* and *C. sepedonicum* [68]. Since then, many other AMPs have been isolated from different plant species. Vilas Boas et al. identified the AMP kalata B1 (cyclotide) from the leaves of the African plant *Oldenlandia affinis.* This peptide showed strong anti-viral activity against human immunodeficiency virus (HIV), having the ability to destroy viral particles and prevent their fusion to the host cell membrane [69]. An antifungal peptide PvD1 (defensin) was successfully purified from *Phaseolus vulgaris* seeds and demonstrated inhibitory growth effects against several yeasts (*Candida albicans, C. parapsilosis, C. guilliermondii, C. tropicalis, Saccharomyces cerevisiae,* and *Kluyveromyces marxiannus*) and fungi (*Rhizoctonia solani, Fusarium solani, F. oxysporum,* and *F. lateritium*) [70]. Next, the AMP snakin-Z was isolated from Ziziphus jujube fruits, showing a great antifungal and antibacterial potential against *Staphylococcus aureus, Escherichia coli, Bacillus subtili,* and *Klebsiella pneumoniae* [71]. The cyclotides Cycloviolacin O2 and Cycloviolacin O8 were isolated from *Viola odorata* [72]. The two compounds demonstrated strong antibacterial activity against various pathogenic bacteria.

It is noteworthy that some parts of the plant do not require extraction. However, in some of these cases, the purification and isolation of these peptides can require more stages [38]. A study conducted by Mandal et al. was able to successfully purify and identify three AMPs (Cn-AMP1, Cn-AMP2, Cn-AMP3) from coconut water (Cocos nucifera L.) [73]. Immediately following the selection of the material, the authors performed dialysis against distilled water with acetic acid addition up to pH 2.0. In order to isolate and purify the AMPs, the resuspended samples were fractionated onto reverse phase chromatography (HPLC). Moreover, these peptides showed antimicrobial activity against *Escherichia coli, Bacillus subtilis, Pseudomonas aeruginosa*, and *Staphylococcus aureus.*

Over the last decades, many efforts have been made in order to use environmentally friendly approaches to extract AMPs from natural sources. Despite all efforts, more studies are needed concerning the development of complete green extraction protocols. Recently, many methodologies of extraction implemented greener steps involving eco-friendly solvents and materials, as previously described. Certain AMPs have already been obtained using some of these techniques. For example, Song et al. identified nine novel AMPs (CHQQEQRP, DENFRKF, EWPEEGQRR, KPPIMPIGKG, KDFPGRR, LGLRSGIILCNV, PRNFQQQLR, QNLNALQPK, and SQEATSPR) from cottonseed protein by enzymatic hydrolysis. These peptides demonstrate a successful inhibitory effect against *Staphylococcus aureus, Escherichia coli, Salmonella* sp., and *Streptococcus* sp. [74]. A recent study conducted by Farhadpour et al. reported the isolation of five vigno cyclotides (vigno 1–5) from *Viola ignobilis* by the microwave-assisted extraction method [75]. Further data have shown that vigno 5 has chemotherapeutic effects on cervical cancer [76].

Over the last three decades, the use of plants and plant cells as system platforms to produce AMPs, also known as molecular farming, has been extensively studied by the biopharmaceutical industry in order to increase the accessibility of recombinant peptides for clinical use [77]. Some of their advantages include a more affordable and rapid process, feasible large-scale production with high yields, the use of simpler manufacturing procedures that reduce extensive purification, and enhanced growth conditions free from toxic contaminants and pathogens. In addition, this expression system offers the major benefit of allowing post-translational modifications to occur, which may be crucial for protein folding and AMPs biological functions. However, regulatory compliance is still a big disadvantage that must be overcome [77]. LFchimera, a chimerical antimicrobial peptide, was recombinantly expressed in the hairy roots of *Nicotiana tabacum* [78]. This peptide exhibited strong antibacterial activity against *Escherichia coli*. Lojewska et al. expressed the recombinant Colicin M peptide in *Nicotiana tabacum* plants and verified its action against *Escherichia coli* and *Klebsiella pneumoniae* [79]. Another study conducted by Patiño-Rodríguez used the same tobacco species as previously reported to express the recombinant broad-spectrum AMP, Protegrin-1 [80]. The authors demonstrated its effectiveness against *Klebsiella pneumoniae, Staphylococcus aureus, Escherichia coli, Mycobacterium bovis*, and the fungal pathogen *Candida albicans.*

#### 5.3.2. Algae

The search for new natural sources of AMPs, turned the scientist’s attention to the marine environment. Throughout evolution, marine organisms have developed a variety of bioactive molecules and strategies to defend themselves from prokaryotic and viral infections [81]. In this context, microalgae and cyanobacteria represent promising resources of molecules with antimicrobial properties [35]. As a result of their flexible metabolism, they are able to adapt to a variety of environmental conditions, including highly competitive environments, and respond to different environmental stresses and nutrient sources. Furthermore, they are exposed to a wide range of predators and microbial pathogens [35]. In response to these stimuli, these microorganisms have developed a wide diversity of antimicrobial compounds that can serve a broad spectrum of applications in the fields of biotechnology, medicine, agriculture, and aquaculture [82].

Cyanobacteria, also known as blue-green algae, are an ancient group of photosynthetic microbes with great ecological importance [83]. These organisms exist as single cells, in pluricellular forms, or as symbiotic partners of other plants and animals [82,83]. Antibacterial peptides isolated from cyanobacteria have been extensively studied and reviewed in the literature over the last decades [35,84,85]. These peptides can be classified into six different categories: cyclic peptides (41.6%), lipopeptides (20.8%), cyclic lipopeptides (16,6%), cyclic depsipeptides (12.5%), linear (4.2%), and depsipeptides (4.2%) [35]. Zainuddin et al. identified four cyclic undecapeptides named lyngbyazothrins A, B, C, and D from the freshwater strain *Lyngbya* sp. as binary mixtures (A/B and C/D). In this study, the authors showed that lyngbyazothrins C/D had antimicrobial activity against both Gram-positive (*Bacillus subtilis*) and Gram-negative (*Escherichia coli, Pseudomonas aeruginosa, Serratia marcesens)* bacteria [84]. Montaser et al., isolated pitipeptolides C-F from the marine cyanobacterium *Lyngbya majuscula* in Piti Bomb Holes, Guam. This study showed that pitipeptolides F was the most potent compound in a disc diffusion assay against *Mycobacterium tuberculosis* [86]. The cyclic undecapeptide, Kawaguchipeptin B, was extracted from the cultured cyanobacterium *Microcystis aeruginosa* (NIES-88) and exhibited antibacterial effects against *Staphylococcus aureus* [87]. Another study conducted by Dussault et al. investigated the effects of ten cyanobacterial isolates on the growth of foodborne pathogens, and concluded that several cyanobacterial peptides (laxaphycins A, B and B3) had antibacterial activity against Gram-positive bacteria [88].

Marine microalgae are unicellular photosynthetic eukaryotic microorganisms that can be found in a variety of aquatic habitats [89]. In the last few decades, these microorganisms have been studied in the search for new high-value molecules with antimicrobial activity that can be used to develop future environmentally friendly antibiotics to combat microbial antibiotic resistance [35,42,82,89,90]. There are a number of advantages to using marine microalgae in drug discovery that include the ease of culture (both on small and large scales), the limited generation time (5–8 h), and the ability to conduct an eco-friendly approach to drug discovery [42,90]. Enzymatic hydrolysis is the most used extraction technique to obtain microalgal AMPs [42]. AMPs derived from microalgae extracts have been predominantly obtained from protein hydrolyzates of different species such as *Chlorella vulgaris*, *Chlorella ellipsoidea*, *Tetradesmus obliquus*, *Navicula incerta*, and *Nannochloropsis oculate* [89]. Several studies have demonstrated that these bioactive microalgal peptides can exert several biological functions, including antioxidant, anticancer, antihypertensive, and antimicrobial properties, with beneficial health effects and potential therapeutic applications [42,89,90]. Guzmán et al. reported three antibacterial peptides from the marine microalgae *Tetraselmis suecica* [89]. In this study, one of the peptides, AQ-1766, which exhibited high activity against Gram-negative bacteria (*E. coli*, *S. typhimurium*, and *P. aeruginosa*) and Gram-positive bacterial strains (*B. cereus*, methicillin-resistant *S. aureus* (MRSA), *L. monocytogenes* and *M. luteus)*, was subjected to a lysine replacement of some residues obtaining six peptides with improved antibacterial activity (AQ-3001, AQ-3002, AQ3369, AQ-3370, AQ-3371, and AQ-3372).

## 6. Therapeutic Use of Green AMPs in DFU

AMPs are a promising alternative for infected wounds since they are active against a wide range of Gram-positive and Gram-negative bacteria. In addition, AMPs can exhibit immunomodulatory and angiogenic properties, stimulate cell proliferation and migration, and accelerate wound healing [2]. Although the AMPs referred above were obtained using green processes and can be relevant as therapeutic option for DFUs (Table 1), some of their inherent steps still need to be improved to develop a completely green protocol. Only a few AMPs are being extracted using more environmentally friendly approaches: Cn-AMPs, SP-1, KLENCNYAVELGK, Lyngbyazothrins mixture C/D, and Laxaphycin A, B, and B3. This may be explained by the recent increase in the awareness for the need of more sustainable and safer processes. AMPs extracted by greener methodologies include the Cn-AMPs: three antimicrobial peptides extracted from coconut water [73]. The extraction of these AMPs has been achieved through simple methodologies that did not use toxic solvents. Nevertheless, reverse phase chromatography (HPLC) was used in their purification stage. In order to reduce the generation of organic waste and accomplish a complete green extraction protocol, this technique could be replaced by new promising and eco-friendly approaches such as MMC, MCSGP, or even SFC [49]. These peptides showed antimicrobial activity against multiple Gram-positive and Gram-negative bacteria and fungi [73]. Interestingly, further studies investigated the promiscuity of these peptides and demonstrated that in addition to their antimicrobial activities, they also present activity against cancerous cells and immunostimulatory effects [91,92,93]. The immunostimulatory effect is a very important property, particularly in diabetes due to the low-grade inflammation, and it is difficult to have a controlled inflammatory phase of wound healing. This is of utmost importance as it will dictate the progression of wound closure [94,95].

Another example is SP-1, an AMP extracted by enzymatic hydrolysis from the cyanobacteria *Spirulina platensis.* The peptide KLENCNYAVELGK was successfully extracted from pepsin hydrolysates derived from *Limnospira sp*. This peptide showed antibacterial activity against *Escherichia coli* and *Staphylococcus aureus* [47]. It also presents antioxidant, anti-hypertensive, anti-diabetes, and anti-obesity activities [45,46]. Most of these peptides have antioxidant properties which are important to promote cell function in wounds under diabetic conditions, particularly, cell migration and proliferation, as well as angiogenic properties [96,97]. Moreover, for a proper wound healing, the overall condition of the patient is very important. The anti-hypertensive and anti-diabetic properties of the peptides will contribute to the control and improvement of the patient’s general condition, and consequently better wound healing. In this sense, peptides that improve the general condition of diabetes could make a better contribution to the treatment of DFUs (Figure 3). In addition, some peptides such as PvD1, cycloviolacin O2, pitipeptolides C-F, and laxaphycins A, B, and B3 have shown to have high cytotoxicity against tumor cells [98,99,100]. This is an important property since diabetic patients present an elevated risk of developing different types of cancers [101,102].

Furthermore, the peptides in the Lyngbyazothrins mixture C/D and Laxaphycin A, B, and B3 obtained by greener methodologies also showed activity against common pathogens found in chronic DFUs [84,88]. The cyclic undecapeptides Lyngbyazothrins C/D showed antimicrobial activity against *Escherichia coli, Pseudomonas aeruginosa, Serratia marcesens,* and *Bacillus subtilis* [84]. Cyanobacterial peptides laxaphycins A, B, and B3 demonstrated antibacterial activity against *Listeria monocytogenes, Bacillus cereus,* and *Staphylococcus aureus* [88].

In addition to the aforementioned peptides, there are others obtained by less eco-friendly methodologies that present interesting properties concerning DFUs treatment. A study reported a peptide known as AQ-1766 was extracted from the marine microalgae *Tetraselmis suecica,* with high activity against *E. coli*, *P. aeruginosa*, MRSA, *L. monocytogenes,* and *M. luteus)* [89]. The fact that this particular AMP possesses high activity against MRSA, an antibiotic-resistant and prevalent pathogen in infected chronic DFUs, makes them an interesting and therapeutic approach for DFUs. Moreover, further research demonstrated that the peptide Snakin-Z also exhibited high antioxidant activity in addition to its antimicrobial properties [103].

Plants and microalgae have been considered sustainable and attractive sources of novel AMPs. Nevertheless, many of the extraction and production processes currently in use still employ hazardous substances [87,88]. The industrial scale-up production of AMPs is still a significant challenge due to highly expensive chemical methodologies [93].

Similarly, the current synthesis methodologies in use are still not completely green, in part due to their use of or the creation of toxic by-products. However, efforts are being made in order to develop more sustainable methodologies. This can be achieved by the replacement of hazardous substances and techniques by more sustainable options. Thus, it is imperative that the current synthesis of synthetic AMPs already identified as potential therapeutic agents for DFUs can be produced using greener techniques.

The potential therapeutic role of most of these peptides for DFU have not yet been investigated. However, due to their multiple properties, it would be important to further study the effects of novel greener AMPs as promising therapeutic agents for non-healing chronic infected wounds.

**Table 1 antibiotics-12-00467-t001:** Antimicrobial peptides extracted from sustainable sources as therapeutic options for DFU.

Source	AMPs/Sequences	Susceptible Species	Other Effects	Ref.
**Plants**				
*Phaseolus vulgaris* seeds	PvD1	*Candida albicans* *Candida parapsilosis* *Candida guilliermondii* *Candida tropicalis* *Saccharomyces cerevisiae*	Activity against tumor cells	[70,98]
Ziziphus jujuba fruits	Snakin-Z	*Staphylococcus aureus* *Escherichia coli* *Bacillus subtili* *Klebsiella pneumoniae*	Antioxidant activity	[71,103]
*Viola odorata*	Cycloviolacin O2	*S. enterica serovar* Typhimurium LT2 *Escherichia coli* *Klebsiella pneumoniae* *Pseudomonas aeruginosa*	Activity against tumor cells	[72,99]
*Cocos nucifera L.*	Cn-AMP1 Cn-AMP2 Cn-AMP3	*Escherichia coli* *Bacillus subtilis* *Pseudomonas aeruginosa* *Staphylococcus aureus*	Activity against tumor cells Immunostimulatory activity	[73]
Cottonseed defatted protein powder	CHQQEQRP DENFRKF EWPEEGQRR KPPIMPIGKG KDFPGRR LGLRSGIILCNV PRNFQQQLR QNLNALQPK SQEATSPR	*Staphylococcus aureus* (ATCC27068) *Escherichia coli* (ATCC25922) *Steptococcus* (CMCC35668) *Salmonella* (CMCC50013)	-	[74]
*Nicotiana tabacum*	LFchimera	*Escherichia coli*	-	[78]
*Nicotiana tabacum*	Colicin M	*Escherichia coli* *Klebsiella pneumoniae*	-	[79]
*Nicotiana tabacum*	Protegrin-1	*Klebsiella pneumoniae* *Staphylococcus aureus* *Escherichia coli* *Mycobacterium bovis* *Candida albicans*	-	[80]
**Microalgae**				
*Spirulina platensis*	SP-1	*Escherichia coli* *Staphylococcus aureus*	Antioxidant, antihypertensive, anti-diabetes, and anti-obesity	[45,46]
*Limnospira maxima*	KLENCNYAVELGK	*Escherichia coli* *Staphylococcus aureus*	-	[47]
*Lyngbya* sp.	Lyngbyazothrins mixture C/D	*Bacillus subtilis* *Escherichia coli* *Pseudomonas aeruginosa* *Serratia marcesens*	-	[84]
*Lyngbya majuscula*	Pitipeptolides C-F	*Mycobacterium tuberculosis*	Activity against tumor cells	[86,100]
*Microcystis aeruginosa* (NIES-88)	Kawaguchipeptin B	*Staphylococcus aureus*	-	[87]
Hawaii and Caribbean collection of cyanobacteria	Laxaphycin A	*Listeria monocytogenes* *Bacillus cereus* *Staphylococcus aureus*	Activity against tumor cells	[88,100]
	Laxaphycin B	*Listeria monocytogenes* *Bacillus cereus* *Staphylococcus aureus*		
	Laxaphycin B3	*Bacillus cereus*		
*Tetraselmis suecica*	AQ-1766 AQ-3001 AQ-3002 AQ3369 AQ-3370 AQ-3371 AQ-3372	*Escherichia coli* *Salmonella typhimurium* *Pseudomonas aeruginosa* *Bacillus cereus* Methicillin-resistant *S. aureus* (MRSA) *Listeria monocytogenes*	-	[89]

## 7. Conclusions and Future Perspectives

Wound healing is a complex biological process which is impaired under diabetes conditions, often leading to chronic non-healing wound infections. The advances in therapeutic approaches to diabetic wound care recognizes the potential of AMPs against infectious pathogens, and also provides other regenerative functions to improve wound healing. Green AMPs represent a promising alternative to conventional antibiotics and could be the answer to help fight antimicrobial resistance. Over the last decades, the raising awareness of climate change and high levels of pollution has increased the consciousness of the need to protect the environment. In this context, the scientific community has been working on the development of novel, greener, and more environmentally friendly methodologies for AMP extraction and synthesis. Plants and microalgae have been considered sustainable, and attractive sources of novel AMPs. Nevertheless, these green sources still have some limitations that need to be overcome. This review underlines the need for further research on new eco-friendly methodologies that allow complete green protocols for AMPs extraction, purification, and synthesis. Moreover, future research on the potential therapeutic role of these green peptides for treating non-healing infected wounds may contribute to the improving the treatment of DFUs.

## Figures and Tables

**Figure 1 antibiotics-12-00467-f001:**
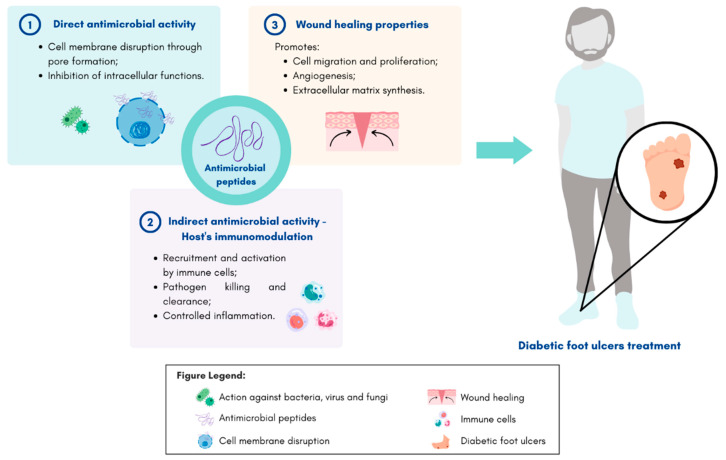
Schematic representation of antimicrobial peptides mechanisms of action and their potential application on diabetic foot ulcer treatment. Figure created in Canva.com (accessed on 24 January 2023).

**Figure 2 antibiotics-12-00467-f002:**
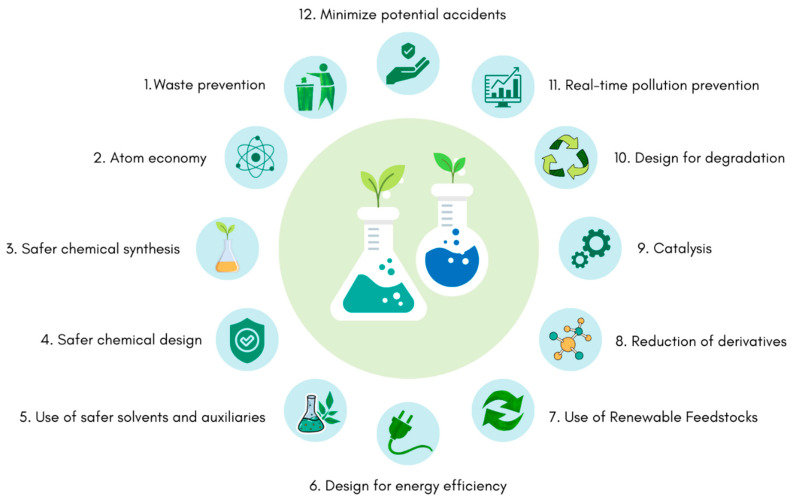
Schematic representation of the twelve principles of green chemistry. Figure adapted from [27] and created in Canva.com (accessed on 24 January 2023).

**Figure 3 antibiotics-12-00467-f003:**
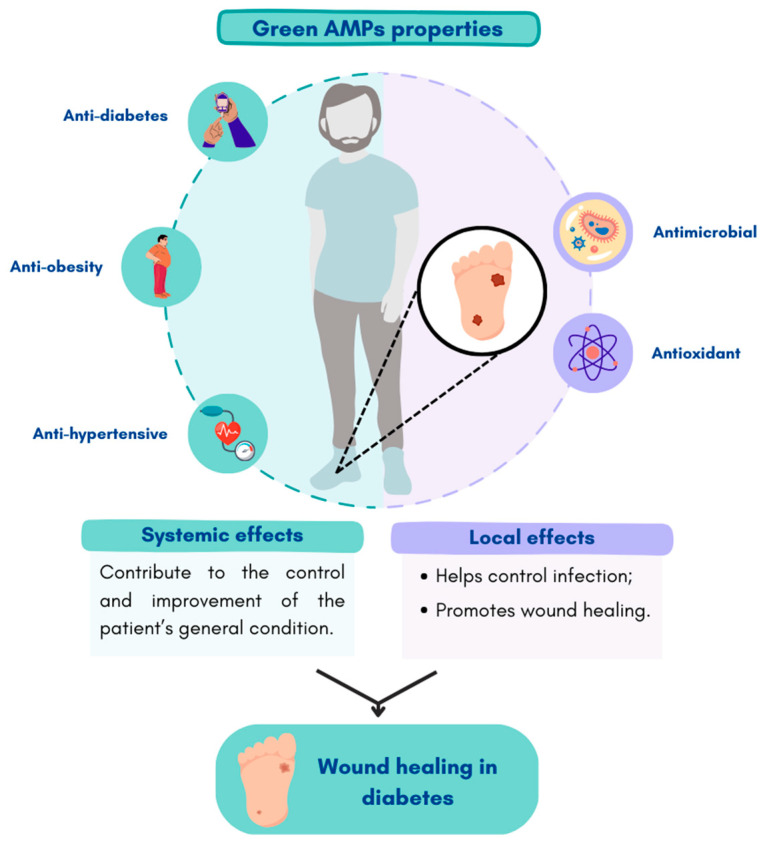
Effects of more environmentally friendly AMPs as potential therapeutic agents in chronic DFUs. Figure created in Canva.com (accessed on 21 February 2023).

## Data Availability

Not applicable.

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
