# Peer review of "Green Antimicrobials as Therapeutic Agents for Diabetic Foot Ulcers"

_antibiotics, 2023, doi:10.3390/antibiotics12030467_

Round 1

Reviewer 1 Report

This is a well-written review on an important and current topic, with a huge number of literature positions. I enjoyed reading it and have only minor comments.

1)      Lines 52-63; Please add information about limitations in the use of AMP (such as cytotoxicity, high production costs, unstable structure) based on the literature.

2)      Line 104; there is no need to repeat the full name along with abbreviation

3)      Figure 1, please consider my suggestions;  please add the information to point 2 “…by immune cells” and add  “Figure legend” above the legend in box.

Author Response

Answers to Reviewer 1

Point 1: Lines 52-63; Please add information about limitations in the use of AMP (such as cytotoxicity, high production costs, unstable structure) based on the literature.

Response 1: Thanks for your suggestion. We added the following information regarding the limitations in the use of AMPs: “Nevertheless, these small peptides can present some limitations such as low structure stability, high cytotoxicity, low hydrosolubility, salt sensitivity, poor selectivity, as well as high production costs [2, 8, 9]. – Lines 58-61”.

Point 2: Line 104; there is no need to repeat the full name along with abbreviation.

Response 2: It was corrected in the text – Line 108 .

Point 3: Figure 1, please consider my suggestions;  please add the information to point 2 “…by immune cells” and add  “Figure legend” above the legend in box

Response 3: Thank you for the suggestion. We changed the figure as suggested by the reviewer.

Reviewer 2 Report

In the review article “ Green Antimicrobials as Therapeutics Agents for Diabetic Foot Ulcers” authors presented examples of antimicrobial peptides (AMPs) derived from green eco-friendly processes with potential antibacterial activity. Authors highlighted the use of AMPs achieved from an eco-friendly method over antibiotics synthesized chemically and identified few AMPs as therapeutic options for Diabetic Foot Ulcers. 

The overall presentation is good though there are few typographical errors that authors need to take care before publication.

Examples:

  1. Line 171- should be host’s
  2. Line 513- cycle 41.6% should be cyclic peptides 41.6%, cycle depsipeptides should be cyclic depsipeptides etc.

Author Response

Answers to Reviewer 2 

Point 1: Line 171- should be host’s

Response 1: Thank you for your feedback. We corrected the error in the text (highlighted) as suggested by the reviewer – Line 175.

Point 2: Line 513- cycle 41.6% should be cyclic peptides 41.6%, cycle depsipeptides should be cyclic depsipeptides etc.

Response 2: This was corrected in the text (highlighted) as the reviewer suggested – Line 535-536.

Reviewer 3 Report

The current manuscript is a very interesting review on green antimicrobials as therapeutic agents for diabetic foot ulcers, especially aspects that concern their synthesis and application. It is a lengthy review, but in a good way, since it covers many important topics in the field. Nevertheless, some alterations should be made before acceptance for publication:

- Section “6. Therapeutic use of green AMPs in DFU” should be completed, since it is the main topic of the review but ends up being only a very small part of the manuscript; for example the paragraph “In addition, the peptides Lyngbyazothrins mixture C/D and Laxaphycin A, B and 584 B3, obtained by greener methodologies, also showed activity against common pathogens 585 found in chronic DFUs [83, 87].” is a quite small description, and hence could be further described, by adding more detailed information about this study’s results, just as the authors did with other studies;

- Additionally, in that same section, images could be added regarding the most promising results of some included studies, if the required permissions for figure reproduction are acquired from the original journals;

- Since the manuscript focuses on green approaches, a paragraph should be added on green analytical chemistry (for example for chromatographic methods), since it is a subject with increasing importance and that could be added to green synthesis in order to make experimental protocols even more environment friendly.

Author Response

Answers to Reviewer 3 

Point 1: Section “6. Therapeutic use of green AMPs in DFU” should be completed, since it is the main topic of the review but ends up being only a very small part of the manuscript; for example the paragraph “In addition, the peptides Lyngbyazothrins mixture C/D and Laxaphycin A, B and 584 B3, obtained by greener methodologies, also showed activity against common pathogens 585 found in chronic DFUs [83, 87].” is a quite small description, and hence could be further described, by adding more detailed information about this study’s results, just as the authors did with other studies;

Response 1: Thank you for your feedback and suggestions. More information was added in section 6:

“Furthermore, the peptides Lyngbyazothrins mixture C/D and Laxaphycin A, B and B3, obtained by greener methodologies, also showed activity against common pathogens found in chronic DFUs [86, 90]. The cyclic undecapeptides Lyngbyazothrins C/D showed antimicrobial activity against Escherichia coli, Pseudomonas aeruginosa, Serratia marcesens and Bacillus subtilis [86]. Cyanobacterial peptides laxaphycins A, B and B3 demonstrated antibacterial activity against Listeria monocytogenes, Bacillus cereus, and Staphylococcus aureus [90].   

In addition to the aforementioned peptides, there are others obtained by less eco-friendly methodologies that present interesting properties concerning DFUs treatment. A study reported a peptide known as AQ-1766, extracted from the marine microalgae Tetraselmis suecica, with high activity against E. coli, P. aeruginosa, MRSA, L. monocytogenes and M. luteus) [91]. The fact that this particular AMP possesses high activity against MRSA, an antibiotic-resistant and prevalent pathogen in infected chronic DFUs, makes them an interesting therapeutic approach for DFUs. Moreover, further research demonstrated that the peptide Snakin-Z also exhibited high antioxidant activity in addition to its antimicrobial properties [106].” – Lines 636-651.

Point 2: Additionally, in that same section, images could be added regarding the most promising results of some included studies, if the required permissions for figure reproduction are acquired from the original journals;

Response 2: We took your suggestion into consideration and added some additional information regarding the most promising green AMPs properties and therapeutic effects in DFU treatment:

“Moreover, for a proper wound healing, the overall condition of the patient is very important. The anti-hypertensive and anti-diabetic properties of the peptides will contribute to the control and improvement of the patient’s general condition, and consequently to better wound healing. In this sense, peptides that improve the general condition of diabetes could make a better contribution to the treatment of DFUs (Figure 3). In addition, some peptides such as PvD1, cycloviolacin O2, pitipeptolides C-F and laxaphycins A, B and B3 have shown to have high cytotoxicity against tumor cells [101-103]. This is an important property, since diabetic patients present an elevated risk of developing different types of cancers [104, 105].” – Lines 603-611.

Some references were added (highlighted) in this section. Also, some information about AMPs “other effects” was added in Table 1 (highlighted).

We also added Figure 3 concerning the “Effects of more environmentally friendly AMPs as potential therapeutic agents in chronic DFUs” – Lines 612-635. 

Point 3: Since the manuscript focuses on green approaches, a paragraph should be added on green analytical chemistry (for example for chromatographic methods), since it is a subject with increasing importance and that could be added to green synthesis in order to make experimental protocols even more environment friendly.

Response 3: Thank you for your suggestion. We agree with the reviewer, and we added the following paragraph:

“Over the last decades, green analytical methodologies have been introduced in order to make experimental protocols even safer and more environmentally benign [65, 66, 67]. In this context, green chromatographic methods have been given increasing importance [65]. The primary goal of green chromatography is to make the process of analysis greener at every stage by replacing existing solvents with greener alternatives, reducing solvent use, and decreasing waste generation [65, 66]. In liquid chromatography (LC), the solvent reduction can be achieved by several strategies like decreasing the column length with small particle size, using ultra-high performance liquid chromatography systems (short columns with reduced internal diameters), microflow and capillary HPLC columns, or even through the use of elevated temperatures [65]. Moreover, several green LC methodologies already implement the use of greener solvents (Brij 35, sodium dodecyl sulfate, and propylene carbonate) as a substitute for toxic solvents [65, 66]. For example, replacing acetonitrile with ethanol (a nontoxic alternative) is a good strategy to improve LC techniques [65]. Also as referred before, several processes can be optimized by the implementation of greener solvents and materials/instruments like resins.”  - Lines 427-441.
